# Evaluation of the Larvicidal Potential of the Essential Oil *Pogostemon cablin* (Blanco) Benth in the Control of *Aedes aegypti*

**DOI:** 10.3390/ph12020053

**Published:** 2019-04-08

**Authors:** Lizandra Lima Santos, Lethicia Barreto Brandão, Rosany Lopes Martins, Erica de Menezes Rabelo, Alex Bruno Lobato Rodrigues, Camila Mendes da Conceição Vieira Araújo, Talita Fernandes Sobral, Allan Kardec Ribeiro Galardo, Sheylla Susan Moreira da Silva de Ameida

**Affiliations:** 1Laboratory of Pharmacognosy and Phytochemistry, Federal University of Amapa, Highway Jucelino Kubistichek, Km 02, Macapa 68.902-280, Brazil; 2Laboratory of Medical Entomology of the Institute of Scientific and Technological Research of the State of Amapa, Highway Jucelino Kubistichek, Km 10, Macapa 68.903-419, Brazil

**Keywords:** Biocide, Patchouli, Oriza, vector control, Lamiaceae

## Abstract

The objective of this work was to collect information on the chemical constituents that demonstrate the larvicidal activity against *Aedes aegypti*, as well as the antioxidant, microbiological, and cytotoxicity potential of the essential oil of *Pogostemon cablin* leaves. The chemical characterization was performed by gas chromatography coupled to mass spectrometer (GC-MS). The larvicidal activity was performed according to the protocol of the World Health Organization. The antioxidant activity was evaluated through the sequestering capacity of 2,2-diphenyl-1-picryl-hydrazine (DPPH). As for the microbiological evaluation, the microdilution technique was used, according to the protocol of the Clinical and Laboratory Standards Institute. The cytotoxic activity was evaluated against the larvae of *Artemia salina*. The species *P. cablin* presented the following compounds: Patchouli alcohol (33.25%), Seyshellene (6.12%), α-bulnesene (4.11%), Pogostol (6.33%), and Norpatchoulenol (5.72%), which was in synergy with the other substances may significantly potentiate the larvicidal action of the species with the LC_50_ of 28.43 μg·mL^−1^. There was no antioxidant activity, however, it presented antimicrobial activity against all bacteria tested with Minimum Inhibitory Concentration (MIC) and Minimum Bactericidal Concentration (MBC) of 62.5 μg·mL^−1^. The species demonstrated significant toxic action with LC_50_ of 24.25 μg·mL^−1^. Therefore, the *P. cablin* species showed significant larvicidal potential, antimicrobial activity, the absence of antioxidant action, and high toxicity.

## 1. Introduction

The use of medicinal plants is an ancient practice used by populations to cure various diseases. This practice is expanding all over the world. Medicinal plants constitute an important and readily available resource found in popular markets, backyards, and areas of native vegetation. Brazil is a living pharmacopeia, due to its size and variety of ecosystems, which offers rich possibilities that can meet the needs of diverse communities, especially those in need [1].

Among these medicinal plants, the study of plant extracts, as well as their essential oils (EO), appears as an expectation to find substances with biocidal activities that can be selected for use in future formulations of a commercial product.

In this context, the species of the Lamiaceae family present huge potential for the obtention of essential oils, which have several biological functions in folk medicine to the treatment of several conditions, such as burns, headache, colic, fever, as well as reports of antiviral activities against influenza, insecticide, insect repellent, antibacterial, and anti-parasitical [2].

The *P. cablin*, popularly known as Oriza and Patchouli, is an evergreen tropical species of plant, originating in Southeast Asia that belongs to the family Lamiaceae, which is currently being extensively cultivated in Malaysia, Indonesia, the Philippines, China, India, Seychelles, and Brazil [3]. The importance of the species of the genus *Pogostemon* is related to allelopathy. Kusuma and Mahfud [4] found 19 compounds of the EO of *P. cablin*. The major compounds were patchouli alcohol (26.2%), δ-guaiene (14.69%), α-guaiene (12.18%), α-gurjunene (11.13%), seychellene (8.42%), viridiflorol (5.93%), β-caryophyllene (4.63%), and β-patchoulene (2.87%).

The composition of the *P. cablin* oil is complex, like those of many other essential oils, but distinct because it is constituted largely by sesquiterpenes. Patchouli alcohol, an oxygenated sesquiterpene, is the major constituent and is primarily responsible for its characteristic aroma.

The oil contains a large number of other hydrocarbons sesquiterpenes such as α, β, σ-patchoulenos, α-bulneseno, α-guaieno, and seicheleno, with structures related to patchouli alcohol and other sesquiterpenes. The accumulation and biosynthesis of patchouli alcohol and related sesquiterpenes in *P. cablin* leaves were studied by their morphology [5].

The substances found in this species have several biological activities described in the literature, such as antioxidant, analgesic, anti-inflammatory, antiplatelet, antithrombotic, aphrodisiac, antidepressant, antimutagenic, antiemetic, fibrinolytic, and cytotoxic [6,7,8,9]. In addition to the insecticide, antibacterial, and antifungal proprieties [6,10,11]. Their recognized entomotoxicity suggest possible applicability as an alternative strategy for the chemical control of *A. aegypti* [12].

*A. aegypti* control has been a major challenge, especially in developing countries, where usually there is the presence of financial resources for the vector control through the implementation of programs that often fail due to aspects related to the lack of garbage collection and intermittent water supply, which are factors that directly influence the traditional *A. aegypti* control methods [13].

In view of the operational and economic difficulties generated by the increasing resistance of mosquitoes to the synthetic insecticides, alternative methods are becoming more prominent, efficient, and cheaper since they are obtained from renewable resources that are rapidly degradable and have several substances which act simultaneously, causing insect resistance to these substances to occur slowly [14].

Thus, in view of the mosquito control challenges and a critical scenario regarding the arboviruses, outlined by the spread of *A. aegypti*-transmitted diseases worldwide, it is essential to adopt specific strategies with greater investments in new methods. Due to the current outbreaks and epidemics of yellow fever, Zika, chikungunya, and dengue, which has victimized thousands of people.

In this context, the present study becomes relevant, since the species *P. cablin* possesses biocidal potential. However, there are still few studies regarding the insecticidal and larvicidal activity of the plant over this vector; as well as sources of preliminary information regarding the search of strategies to reduce the contact of the mosquito with humans, such as the insertion of herbal medicines into the public health system, which would assist in the quality of the health primary care as vector control.

It is also worth noting that for the formulation of a commercial product, it is important to carry out a study of biological activities such as antioxidant, microbiological, and cytotoxic as a complement to prove the phytotherapeutic potential of the species.

Therefore, the present study aimed to collect information on chemical constituents that demonstrate the larvicidal, antioxidant, antimicrobial, and cytotoxicity activities of the EO of the leaves of *P. cablin*.

## 2. Material and Methods

### 2.1. Plant Material

The species *P. cablin* was collected in the district of Fazendinha (00°02′23″ S and 51°06′29″ O) in the Macapa Municipality, in the Amapa State, Brazil. For the taxonomic identification, samples of the species were deposited in the Amapaense Herbarium (HAMAB) of the Institute of Scientific and Technological Research of the State of Amapa (IEPA) under the registration number 019183.

### 2.2. Essential Oils

The leaves of *P. cablin* were dehydrated in an incubator with air circulation at 36 °C, and after being dried, they were crushed in an electric mill. The EO was extracted by hydro-distillation at 100 °C in Clevenger-type apparatus for three hours [15].

### 2.3. Chemical Analysis

The chemical composition of the EO was determined by gas chromatography coupled to mass spectrometer (GC-MS), using a model GCMS-QP 5050A, manufactured by Shimadzu company (Kyoto, Japan), under the following conditions: DB-5HT column of the brand J and W Scientific, with length of 30 m, diameter of 0.32 mm, film thickness 0.10 μm, and nitrogen as carrier gas, according to Martins et al. [16].

The apparatus operated under internal column pressure of 56.7 kPa, split ratio 1:20, the gas flow in the column was of 1.0 mL·min^−1^ (210 °C), injector temperature of 220 °C, and in the spectrum of the mass of 240 °C. The initial temperature of the column was 60 °C with an increase of 3 °C·min^−1^, until reaching 240 °C, kept constant for 30 min.

The mass spectrometer was programmed to perform readings in a range of 29 to 400 Da, at intervals of 0.5 s, with ionization energy of 70 eV. 1 μL of each sample with a concentration of 10.000 ppm dissolved in hexane was injected.

The identification of individual components was based on the comparison of their retention index (RI) and mass spectra with the literature [17]. The RI was calculated relative to a number of n-alkanes (C8-C40, Sigma-Aldrich, St. Louis, MO, USA) using the Van Den Dool and Kratz equation [18].

### 2.4. Larvicidal Activity

The larvae of *A. aegypti* employed in the larvicidal test were from a colony kept in the insectary of the Medical Entomology Laboratory of the Institute of Scientific and Technological Research of the State of Amapa (IEPA). The biological assays were conducted under controlled climatic conditions with a temperature of 25 ± 2 °C, relative humidity of 75 ± 5% and a photoperiod of 12 h. The methodology adopted followed the World Health Organization standard protocol [19], with a slight modification regarding the test vessel.

The EO of *P. cablin* (0.09 g) was dissolved in 85.5 mL of distilled water and 4.5 mL of Tween 80; and for the negative control, it was used respectively Tween 80 with distilled water (1%), and the larvicidal esbiothrin as the positive control.

After the preliminary tests, the aqueous solution was diluted in the following concentrations: 100, 60, 40, 20, 10, and 1 μg·mL^−1^. Each concentration was tested in triplicate, and 25 larvae of the *A. aegypti* mosquito in the 3rd young stage (L3) were used. They were pipetted into a 100 mL beaker containing distilled water, then they were transferred into the test vessels, minimizing the time between the preparation of the first and last samples. During the experiment, the average water temperature was 25 °C. After 24 and 48 h, the dead larvae were counted, being considered as such, all those unable to reach the surface.

### 2.5. Determination of Antioxidant Activity

The evaluation of the antioxidant activity was based on the sequestering ability of 2,2-diphenyl-1-picrylhydrazyl (DPPH), as proposed by Chen, Berlin and Froldi [20], and Lopez-Lutz et al. [21] with modifications. The antioxidant activity was calculated [22] as follows:
(%AA) = 100 − {[(Abs_sample_ − Abs_white_)100]/Abs_control_}

%AA—percentage of antioxidant activityAbs_sample_—Sample absorbanceAbs_white_— White absorbanceAbs_control_—Control absorbance

A methanolic solution of DPPH at the concentration of 40 μg·mL^−1^ was prepared. The EO was diluted in methanol at the following concentrations: 7.81; 15.62; 31.25; 62.5; 125 and 250 μg·mL^−1^. The antioxidant activity evaluation was made in triplicate with a volume of 0.3 mL of EO solution per tube, added to 2.7 mL of the DPPH solution. In parallel, the negative control of each concentration was prepared. For positive control, ascorbic acid was used under the same conditions of EO preparation. After 30 min of incubation at room temperature and protected from light, the spectrophotometer, of the manufacturer Biospectro, model SP-22 (Curitiba, BRA) was measured at wavelength 517 nm in a quartz cuvette.

### 2.6. Antimicrobial Activity

#### 2.6.1. Bacterial Strains and Culture Conditions

Two gram-negative bacteria (*Pseudomonas aeruginosa* ATCC 25922, *Escherichia coli* ATCC 8789) and a gram-positive bacterium (*Staphylococcus aureus* ATCC 25922) were used to test the antimicrobial activity of the EO of *P. cablin* leaves.

From a stock culture in BHI (Brain Heart Infusion) with 20% glycerol stored at −80 °C the activation of each microorganism as performed by transferring 50 μL of this culture into 5 mL of sterile BHI broth followed by incubation for 24 h at 37 °C.

#### 2.6.2. Determination of Minimum Inhibitory Concentration (MIC) and Minimum Bactericidal Concentration (MBC)

The determination of the MIC and the MCB was performed using the microplate dilution technique (96 wells) according to the protocol established by the Clinical and Laboratory Standards Institute [23], with adaptations.

Initially, the bacteria were reactivated in BHI broth, for 18 h at 37 °C. After bacterial growth, a 0.9% saline an inoculum adjusted to the McFarland 0.5 scale was prepared for each microorganism, and subsequently diluted in BHI and tested at 2 × 106 CFU·mL^−1^.

For the determination of MIC, the EO was diluted in Dimethyl sulfoxide (2% DMSO). The first well column of the plate was filled with 0.2 mL of the EO at the concentration of 2000 μg·mL^−1^, the other wells were filled with 0.1 mL of 0.9% NaCl. Subsequently, base two serial dilutions were performed in the ratio of 1:2 to 1:128 dilution in a final volume of 0.1 mL. Cells (2 × 106 CFU·mL^−1^) with 0.1 mL adjusted according to the previous item were added to each well, resulting in a final volume of 0.2 mL. Control of the culture environment, EO control, and negative control (DMSO 2%) were performed. For positive control, amoxicillin (0.5 μg·mL^−1^) was used. The experiments were carried out in triplicates. The microplates were incubated at 37 °C for 24 h, after this time, the plates were read in an ELISA reader (OD 630 nm).

The mean values of OD 630 nm obtained were plotted against essential oil (test substance) concentrations after subtracting the mean OD 630 nm values of the diluted oil in BHI medium with 2% DMSO (turbidity control). MIC was considered to be the lowest concentration of the test substance in which there was no significant bacterial growth compared to the negative control (comparison between the values of D.O.630 by the Bonferroni test with 99% confidence interval).

The MCB was determined based on the results obtained in the MIC test. Microplate wells were spread in Müller-Hinton agar and incubated at 37 °C for 24 h. The MBC was established as the lowest concentration of EO capable of completely inhibiting microbial growth in Petri dishes after 24–48 h of growth.

### 2.7. Cytotoxic Activity

The cytotoxicity of the EO was evaluated against larvae of *Artemia saline* [24,25] with adaptations. A solution of 250 mL of synthetic sea salt at 35 g·L^−1^ was prepared, in which 25 mg of *A. salina* Leach eggs were exposed to artificial lighting within 24 h to hatch the larvae (nauplii). The nauplii were then separated and placed in a dark environment at room temperature for a further 24 h to reach the methanauplii stage.

A stock solution was prepared to contain 0.06 g of the EO, 28.5 mL of the solution of synthetic sea salt and 1.5 mL of Tween 80 added to facilitate solubilization. For the negative control, it was used respectively Tween 80 with solution saline (5%), and the (K_2_Cr_2_O_7_) Potassium dichromate (1%) as the positive control. Later, at the end of the dark period, they were selected and divided into 7 groups with 10 methanauplii in each test tube. In each group it was added aliquots of the stock solution of 100, 75, 50, 25, and 2.5 μL and completed the volume to 5 mL with a solution of synthetic sea salt, obtaining solutions with final concentrations of 40, 30, 20, 10, and 1 μg·mL^−1^ in triplicates.

### 2.8. Statistical Analysis

The results obtained from the bioassays were expressed through Averages and Standard Deviation, categorized in Microsoft Excel (Version 2010 for Windows, Redmond, WA, USA). The graphs were built on GraphPad Prism software (Version 6.0 for Windows, San Diego, CA, USA). Significant differences between treatments were assessed using the ANOVA test One criterion and the Tukey test using the BioEstat program (Version 5.0 for Windows, Belem, BRA). The LC_50_ values were determined in the PROBIT regression, through the SPSS statistical program (Version 21 for Windows, Chicago, IL, USA). Differences that presented probability levels less than or equal to 5% (*p* ≤ 0.05) were considered statistically significant.

## 3. Results

### 3.1. Chemical Analysis

For GC-MS analysis of EO of *P. cablin*, it was possible to identify 29 compounds divided between sesquiterpenes and oxygenated terpenes, according to Table 1.

Among the compounds identified, the following constituents were found: Seyshellene (6.12%), α-bulnesene (4.11%), Norpatchoulenol (5.72%), Pogostol (6.33%), and Patchouli alcohol (33.25%), shown in Figure 1. Mass spectra are available in the Appendix A.

### 3.2. Larvicidal Activity

The *P. cablin* EO presented excellent larval mortality at low concentrations. Additionally, this is shown in Table 2 regarding a period of 24 and 48 h.

Essential oils with LC_50_ below 100 ppm (100 μg·mL^−1^) are considered good agents with larvicidal potential [26]. According to the Table 2, the results demonstrated that the EO of *P. cablin* presents a significant larvicidal effect with LC_50_ of 28.43 μg·mL^−1^, *p*-value < 0.05 and coefficient of determination (R^2^) of 0.0963 in 24 h, result with a high value when compared to the standard larvicidal Esbiothrin, with LC_50_ of 0.0034 μg·mL^−1^.

### 3.3. Antioxidant Activity

The correlation between the antioxidant activity (%) and the EO concentration presented a high IC_50_ value with 329.81 μg·mL^−1^ when compared to the standard Ascorbic acid (vitamin C) with IC_50_ of 16.71 μg·mL^−1^. The results showed that the species under study did not present antioxidant activity, since the IC_50_ of the correlation between antioxidant activity (%) and the EO concentration was higher than that of the positive control, besides the DPPH consumption being smaller than 50% in all the concentrations tested.

### 3.4. Antimicrobial Activity

In the evaluation of the antimicrobial activity, the microorganisms *P. aeruginosa* (ATCC 25922), *S. aureus* (ATCC 25922) and *E. coli* (ATCC 8789) were used for the experiments. The results were expressed as Minimum Inhibitory Concentration (MIC) and Minimum Bactericidal Concentration (MBC), according to Figure 2.

The vitro antimicrobial activity, assays demonstrated that the bacteria *S. aureus*, *P. aeruginosa*, and *E. coli* bacteria were susceptible to the EO of *P. cablin* with a MIC and MBC the concentration of 62.5 μg·mL^−1^, which corresponds to a high value when compared to Amoxiline, which presented MIC and MBC at the concentration of 0.048 μg·mL^−1^.

### 3.5. Cytotoxic Activity

Table 3 shows the average mortality rates obtained after a 24 h period of exposure to the EO of *P. cablin*. against *A. salina*.

The data in Table 3 show that the EO of *P. cablin* presents a LC_50_ 24.25 μg·mL^−1^, *p*-value < 0.05 and coefficient of determination R^2^ of 0.902. This value is above the toxicity standard of potassium dichromate with LC_50_ of 12.60 μg·mL^−1^, however, the EO of *P. cablin* has a high toxic action, because, according to Martins et al. [16], pure substances extracted from plants are considered toxic when LC_50_ < 100 μg·mL^−1^, and nontoxic with LC_50_ > 1000 μg·mL^−1^.

## 4. Discussion

The study for the development of herbal remedies with larvicidal action against *A. aegypti* is recent, beginning in the 1980s, in order to isolate and characterize such bioactive substances. Most of the studies are carried out from raw extracts and essential oils. In most of these cases, the compound responsible for the activity is not known because its action occurs more effectively when grouped with other substances.

Many herbal products have active compounds, which act synergistically or in isolation, having characteristics that can be effective for the control and monitoring of mosquito populations [27].

Out of the compounds found in the chemical analysis of *P. cablin*, patchouli alcohol was the major constituent with a relative percentage of 33.25%, followed by the compounds Seyshellene (6.12%), α-bulnesene (4.11%), Norpatchoulenol (5.72%), and Pogostol (6.33%). An approximate amount of the main substance of the present study was found in a study by Albuquerque et al. [28], which identified patchouli alcohol as the predominant compound in its oil, with a relative percentage of 36.60%.

The amount of patchouli alcohol found in this species cultivated in the state of Amapá-Brazil, differs from many studies in the literature that have also highlighted patchouli alcohol as the major compound [4,5,6,8,9,28,29], due to several factors, including the genetic factor, climate, soil conditions, cultural management, and nutrition, where the last one is considered the most important, since deficiency or excess of nutrients may interfere with the production of the active substances in the species [30].

The insecticidal potential of essential oils is related to chemical constituents [31]. Since essential oils have a complex mixture of several compounds, the exact definition of those that act in the chemical control of immatures becomes a complex task because the biological effects may be a result of the major component or the synergistic action of these constituents [32].

However, there are reports in the literature indicating that sesquiterpenes have several bioactivities [33,34], such as patchouli alcohol present insecticidal activity against *A. aegypti* larvae, as evidenced by the study conducted by Autran et al. [35], which analyzed the essential oils of leaves, stems, and inflorescences of *Piper marginatum*, finding 40 chemical components, and among them the patchouli alcohol was the major component that presented potent larvicidal activity with LC_50_ of 20 ppm, similar to the results of the present study, in which the EO of *P. cablin* had a significant larvicidal effect against larvae of *A. aegypti* with LC_50_ of 28.43 μg·mL^−1^, and production of patchouli alcohol as the major constituent also occurs. Different from this study, Paulraj [36] found a high LC_50_, with a value of 200 in the larvicidal analysis of the EO of *P. cablin* against *A. aegypti*.

As for the antioxidant analysis, Tohidi et al. [37] emphasized that the samples with high antioxidant potential have the capacity of sequestering free radicals and possess low IC_50_ values. Thus, a small amount of sample is capable of decreasing the initial concentration of the DPPH radical by 50%, i.e., inhibiting the radical oxidation by 50%. Thus, from the results observed in this study, there was no antioxidant activity, since the correlation of the IC_50_ and between the antioxidant activity (%) and EO concentration was 329.81 μg·mL^−1^, which was higher when compared to the standard ascorbic acid (vitamin C) with IC_50_ of 16.71 μg·mL^−1^.

Studies related to the antioxidant activity performed with essential oils of the Lamiaceae family, including the *P. cabin* species, also showed a high IC_50_ value with 225.7 μg·mL^−1^ and did not present a significant antioxidant capacity through analysis by the DPPH method, data similar to the results of this study [38].

For Beatović et al. [39], the antioxidant capacity of the EO is related to its major compounds. The content of these phytochemicals in the plants is largely influenced by genetic factors, environmental conditions, besides the degree of maturation and plant variety, among others. It is also emphasized that the antioxidant activity is influenced by the lipid substrate used in the test, the solvent and the extraction technique were employed.

In relation to the antimicrobial activity, the EO of *P. cablin* presented antimicrobial action with MIC and MBC at the concentration of 62.5 μg·mL^−1^. Studies conducted by Adhavan et al. [40] on the antimicrobial activity of nanoemulsions from three different genera of *Pogostemon*, showed that the essential oil of *P. cablin* presented antimicrobial activity at the concentration of 12.5 mg·mL^−1^ for the *S. aureus* bacterium, a result above the concentration found in this one study.

The antimicrobial evaluation of EO of *P. cablin* in the studies conducted by Yang et al. [41] also showed good activity, with MIC at concentrations of 4.0, 5.0, and 4.0 mg·mL^−1^ and MBC at concentrations of 2.0, 10.0 and 6.5 mg·mL^−1^ against *E. coli*, *P. aeruginosa,* and *S. aureus*, respectively.

Therefore, the in vitro antimicrobial susceptibility testing of *P. cablin* essential oil has demonstrated a significant antimicrobial potential, as highlighted by Pattnaik et al. [42] that observed the inhibition of 20 bacteria promoted by the patchouli oil. This biological activity is directly related to the qualitative and quantitative composition of EO of *P. cablin*, which consists of more than 24 sesquiterpenes, in addition to mixtures of different mono, sesqui, and di-terpene compounds [9].

The preliminary toxicological bioassay with *A. salina* allows to verify if the effects that a compound produces in these microcrustaceans are applicable to the human [43]. In this study, the EO of *P. cablin* presented significant toxic action (LC_50_ of 24.25 μg·mL^−1^). Similar data were found by Powers et al. [44] who studied the toxicity the EO of *P. cablin* to human breast tumor cell lines, in which in one of tumors found high toxicity with LC_50_ of 25.0 μg·mL^−1^.

The significant toxicity of the EO of *P. cablin* is attributed to the chemical compounds that constitute it [45], that is, its oxygenated compounds (monoterpenes and sesquiterpenes) such as patchouli alcohol, the main chemical component of the species. However, the essential oil may be more toxic than its isolated compounds, because of the synergistic effect among its constituents, which increases its effectiveness [46].

## 5. Conclusions

The chemical composition of the essential oil of *P. cablin* indicated the presence of 29 substances. The main components identified were patchouli alcohol (33.25%), Seyshellene (6.12%), α-bulnesene (4.11%), Pogostol (6.33%), and Norpatchoulenol (5.72%).

The EO of *P. cablin* showed significant larvicidal potential and could be used to control *A. aegypti* mosquito larvae, suggesting the study with other vectors.

As for the antioxidant evaluation, there was no evidence of antioxidant activity by the DPPH method when compared to the vitamin C standard.

In the preliminary assessment of toxicity with *A. salina*, the EO presented significant toxic action, however, more comprehensive studies should be carried out to prove the toxicity of this species to humans and the environment.

The antimicrobial activity showed that the essential oil of *P. cablin* presented antimicrobial action with a MIC and MBC at the concentration of 62.5 μg·mL^−1^ against all the bacteria tested.

Thus, according to the literature consulted, the results of the biological activities of this study may be associated with the major compound of the species: Patchouli alcohol. However, the diversity of the chemical composition of this species is the most relevant factor associated with the biological actions proven by research that directs and supports the action of the major compounds alone or in combination (synergy).

The data show the relevance of the bioassays as a trial tool for the potential of the biological of the *P. cablin* species, as well as the importance of these as a source of biocidal compounds.

## Figures and Tables

**Figure 1 pharmaceuticals-12-00053-f001:**
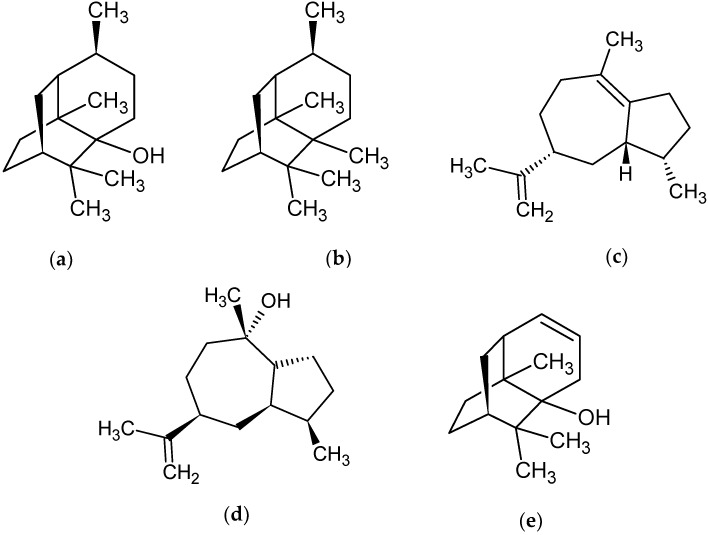
Molecular structure of the major compounds of *P. cablin* essential oil. (**a**) Patchouli alcohol (33.25%), (**b**) Seyshellene (6.12%), (**c**) α-bulnesene (4.11%), (**d**) Pogostol (6.33%), and (**e**) Norpatchoulenol (5.72%).

**Figure 2 pharmaceuticals-12-00053-f002:**
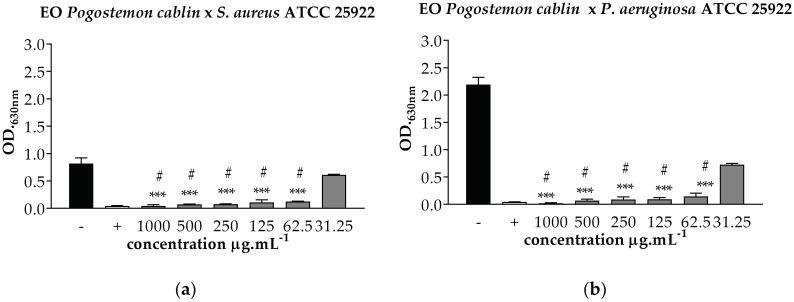
MIC and MBC of the EO of *P. cablin* against (**a**) *S. aureus*, (**b**) *P. aeruginosa* and (**c**) *E. coli.* Source: Own author. Substance test (
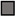
), BHI with 2% DMSO (
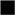
), and Amoxiline (
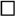
). *** *p* < 0.001 statistically significant in relation to the negative control, # *p* < 0.001 statistically significant in relation to the positive control.

**Table 1 pharmaceuticals-12-00053-t001:** Substances identified by GC-MS analysis of *P. cablin* essential oil.

Peak *	RT (min)	RI	Compounds	Relative Percentage
1	23.942	1382	β-patchoulene	0.46%
2	24.183	1387	β-elemene	0.88%
3	25.275	1412	Cycloseychellene	0.65%
4	25.467	1417	(*E*)-caryophyllene	0.65%
5	26.217	1434	α-guaiene	2.99%
6	26.817	1448	Seychellene	6.12%
7	27.042	1453	α-humulene	0.42%
8	27.292	1459	α-patchoulene	3.59%
9	27.558	1465	9-epi-(*E*)-caryophyllene	1.24%
10	27.875	1473	β-chamigrene	0.14%
11	28.450	1486	β-selinene	0.19%
12	28.750	1493	Aciphyllene	0.54%
13	28.875	1496	Viridiflorene	0.57%
14	29.042	1500	α-bulnesene	4.11%
15	29.708	1516	7-epi-α-selinene	0.15%
16	31.042	1549	Elemol	0.16%
17	31.942	1571	Norpatchoulenol	5.72%
18	32.342	1580	Caryophyllene oxide	3.86%
19	32.458	1583	Globulol	1.79%
20	32.950	1595	Fokienol	0.67%
21	33.442	1607	Humulene epoxide II	0.72%
22	34.267	1629	Junenol	1.87%
23	34.717	1640	*allo*-aromandendrene epoxide	1.82%
24	35.742	1666	Pogostol	6.33%
25	36.308	1681	Patchouli alcohol	33.25%
26	37.083	1701	Thujopsenal	2.06%
27	37.758	1719	*Z*-α-atlantone	1.44%
28	38.225	1732	Isobicyclogermacrenal	0.77%
29	39.925	1777	Squamulosone	0.97%
Compounds identified				84.3%
Unidentified compounds				15.87%

RI = Retention Index of Van den Dool and Kratz (1963); RT = Retention Time; * The peaks are numbered according to the chromatogram available in the Appendix A.

**Table 2 pharmaceuticals-12-00053-t002:** Percentage of mortality (%) of *A. aegypti* larvae in different concentrations of essential oil of *P. cablin* in two periods.

Concentrations	Larvicidal Activity (%)	
(µg·mL^−1^)	24 h	48 h
Control (−)	0.0	0.0
20	38.0	70.66
40	52.0	89.33
60	92.0	97.33 ^a^
80	94.66 ^a^	98.66 ^a^
100	96.0 ^a^	98.66 ^a^
LC_50_ (EO)	28.43 μg·mL^−1^	
LC_50_ (Control +)	0.0034 μg·mL^−1^	

^a^ Statistically significant in relation to the positive control.

**Table 3 pharmaceuticals-12-00053-t003:** Mortality percentage of *A. salina* larvae due to exposure to the essential oil of *P. cablin*.

Concentrations (µg·mL^−^^1^)	Mortality (%)
Control negative	0.0% ^a^
1	3.3% ^a^
10	20.0% ^a^
20	45.0% ^b^
30	56.6% ^c^
40	76.6% ^d^
LC_50_(EO)	24.25 μg·mL^−1^
LC_50_ (K_2_Cr_2_O_7_)	12.60 μg·mL^−1^

Different letters indicate that there was a significant difference between the concentrations (*p* < 0.05).

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
