# Peer review of "Evaluation of the Larvicidal Potential of the Essential Oil Pogostemon cablin (Blanco) Benth in the Control of Aedes aegypti"

_pharmaceuticals, 2019, doi:10.3390/ph12020053_

Round 1

Reviewer 1 Report

The paper describes a comprehensive investigation of the essential oils on the larvicidal activity against

 Aedes aegypti, antioxidant, microbiological and cytotoxic activities of the essential oil of Pogostemon cablin (Blanco) Benth.

·         The title is not ideal.

·         The overall paper needs to be proof-read. There are some mistakes which need to be corrected, such as IC50, gram negative

·         The digits of all the values need to be standardized.

·         Figure 1, please indicate the absolute configurations for all compounds if available. It is not necessary to number the carbon atoms.

·         I recommend adding the following publications as important backgrounds.  

·         If possible, one figure of GCMS profile needs to be uploaded. It needs to show the components that are not yet identified.   

1.       Absolute Configurations and Bioactivities of Guaiane-Type Sesquiterpenoids Isolated from Pogostemon cablin.Zhou QM, Chen MH, Li XH, Peng C, Lin DS, Li XN, He Y, Xiong L.J Nat Prod. 2018 Sep 28;81(9):1919-1927.

2.       Lethal Effect and Behavioral Responses of Leaf-Cutting Ants to Essential Oil of Pogostemon cablin (Lamiaceae) and Its Nanoformulation. Rocha AG, Oliveira BMS, Melo CR, Sampaio TS, Blank AF, Lima AD, Nunes RS, Araújo APA, Cristaldo PF, Bacci L. Neotrop Entomol. 2018 Dec;47(6):769-779.

3.       Pocahemiketals A and B, two new hemiketals with unprecedented sesquiterpenoid skeletons from Pogostemon cablin. Zhu H, Zhou QM, Peng C, Chen MH, Li XN, Lin DS, Xiong L. Fitoterapia. 2017,120:67-71.

4.       Insecticidal and repellence activity of the essential oil of Pogostemon cablin against urban ants species. Albuquerque EL, Lima JK, Souza FH, Silva IM, Santos AA, Araújo AP, Blank AF, Lima RN, Alves PB, Bacci L. Acta Trop. 2013,127(3):181-6.

Author Response

Answer letter

·      Reviewer: The title is not ideal.

Authors: The title was corrected for "Evaluation of the larvicidal potential of the essential oil Pogostemon cablin (White) Benth in the control of Aedes aegypti".

·      Reviewer:  The overall paper needs to be proof-read. There are some mistakes which need to be corrected, such as IC50, gram negative.

Authors: The article was revised and the errors were corrected as LC50 and IC50 for LC50 and IC50, as well as other formatting errors that were also corrected.

·      Reviewer: The digits of the values to be standardized.

Authors: The digits of the values were standardized according to correction request.

·      Reviewer: Figure 1, please indicate the absolute configurations for all compounds if available. It is not necessary to number the carbon atoms.

Authors: The tags were inserted for all components of Figure 1. The number the carbon atoms was removed.

·      Reviewer: I recommend adding the following publications as important backgrounds.

1.       Absolute Configurations and Bioactivities of Guaiane-Type Sesquiterpenoids Isolated from Pogostemon cablin.Zhou QM, Chen MH, Li XH, Peng C, Lin DS, Li XN, He Y, Xiong L.J Nat Prod. 2018 Sep 28;81(9):1919-1927.

2.       Lethal Effect and Behavioral Responses of Leaf-Cutting Ants to Essential Oil ofPogostemon cablin (Lamiaceae) and Its Nanoformulation. Rocha AG, Oliveira BMS, Melo CR, Sampaio TS, Blank AF, Lima AD, Nunes RS, Araújo APA, Cristaldo PF, Bacci L. Neotrop Entomol. 2018 Dec;47(6):769-779.

3.       Pocahemiketals A and B, two new hemiketals with unprecedented sesquiterpenoid skeletons from Pogostemon cablin. Zhu H, Zhou QM, Peng C, Chen MH, Li XN, Lin DS, Xiong L. Fitoterapia. 2017,120:67-71.

4.       Insecticidal and repellence activity of the essential oil of Pogostemon cablin against urban ants species. Albuquerque EL, Lima JK, Souza FH, Silva IM, Santos AA, Araújo AP, Blank AF, Lima RN, Alves PB, Bacci L. Acta Trop. 2013,127(3):181-6.

Authors:  All references have been inserted in.

·         Reviewer: If possible, one figure of GCMS profile needs to be uploaded. It needs to show the components that are not yet identified. 

Author: Table 1 (in the article) includes all compounds that could be identified, with 84.13% of the total present in the sample. Thus, 15.87% was the percentage of unidentified compounds through GC MS analysis.

In the attached file, the Table 1 shows the identified and unidentified compounds, totaling 100%. The respective peaks indicated in the table refer to those shown in the chromatogram of figure 1.

NOTE: In the attached file is the revised article, the response to the reviewer and the supplementary material

Reviewer 2 Report

The objectives of the article by L.L. Santos et al., are the study of the larvicidal (against Aedes aegypti) antioxidant (DPPH), antimicrobial (against P. aeruginosa, E. coli and S. aureus) and cytotoxic action of the Pogostemon cablin (Blanco) Benth essential oil. Based on the current scientific efforts for a comprehensive study of the medicinal plants usefulness, the article is of importance to the scientific community. However, there are a lot of concerns and therefore the manuscript needs a major revision.

-The title should be more precise.  

-The manuscript needs to be reviewed again by the authors for typing and sense errors

-Some crucial information regarding the various methods is not presented (i.e. how the MIC was estimated by the Optical Density of the microplates used).

-Reference controls should always be employed in proper amounts and compared accordingly with the results (for example the use of 1000 μg/μL amoxicillin in antibacterial estimation).

-Statistical analyses were performed with the use of four different software packages (MS excel, GraphPab, BioEstat and SPSS) and are not presented properly in the Statistical analysis section of M&M. Various significance levels are interchanging in the description of the analysis and in the results.

-It would be better if at least two different methods have been used in the estimation of antioxidant activity. 

-Table 1 needs to be rechecked. For example the relative concentration of patchouli alcohol is 3.25% or 33.25%?

-Section 3.3 should be “Antioxidant activity” and not “Larvicidal activity”.

-Section 3.4 should be “antimicrobial“ and not “microbiological” activity.

-Figure 2 is not helping someone to comprehend the antimicrobial activity of the essential oils.

-Line 227. Table 6 is missing

-Lines 263-293 of the discussion are repeated again in lines 294-324.

-The discussion should focus in the comparison of the results from this study to those already published.

-Line 403. Your results are not supporting this conclusion.

Author Response

Answer letter

·      Reviewer: The title should be more precise. 

Authors: The title was corrected for "Evaluation of the larvicidal potential of the essential oil Pogostemon cablin (White) Benth in the control of Aedes aegypti".

·      Reviewer: The manuscript needs to be reviewed again by the authors for typing and sense errors.

Authors: The manuscript was revised and the formatting errors were corrected.

·         Reviewer: Some crucial information regarding the various methods is not presented (i.e. how the MIC was estimated by the Optical Density of the microplates used).

Authors: The mean values of OD 630 nm obtained were plotted against essential oil (test substance) concentrations after subtracting the mean OD 630 nm values of the diluted oil in BHI medium with 2% DMSO (turbidity control). MIC was considered to be the lowest concentration of the test substance in which there was no significant bacterial growth compared to the negative control (comparison between the values of D.O.630 by the Bonferroni test with 99% confidence interval).

The paragraph above was inserted in the line 175.

·         Reviewer: Reference controls should always be employed in proper amounts and compared accordingly with the results (for example the use of 1000 μg/μL amoxicillin in antibacterial estimation).

Authors: According to guidelines established by the Clinical and Laboratory Standards Institute (CLSI, 2018) and followed internationally, the minimum inhibitory concentration to ensure the effectiveness of amoxicillin against tested species should be less than or equal to 0.5 μg.mL-1.  In line 172 of the revised article, the amoxillin concentration was corrected.

·         Reviewer: Statistical analyses were performed with the use of four different software packages (MS excel, GraphPab, BioEstat and SPSS) and are not presented properly in the Statistical analysis section of M&M. Various significance levels are interchanging in the description of the analysis and in the results.

Authors: The four software packages were entered in session 2.8 statistical analysis. Regarding the levels of significance, in the antimicrobial tests and results, a significance level of 99% was considered, in the other tests the significance of 95% was used, since the analyzes were performed in different programs.

·         Reviewer: It would be better if at least two different methods have been used in the estimation of antioxidant activity. 

Authors: However, it was not possible to perform another test due to the short-term response of the manuscript. Therefore, new antioxidant tests will be performed later.

·         Reviewer:  Table 1 needs to be rechecked. For example the relative concentration of patchouli alcohol is 3.25% or 33.25%?

Authors: It's 33.25. The table has been corrected.

·         Reviewer: Section 3.3 should be “Antioxidant activity” and not “Larvicidal activity”.

Authors: This item has been corrected.

·         Reviewer: Section 3.4 should be “antimicrobial“ and not “microbiological” activity.

Authors: This item has been corrected.

·         Reviewer: Figure 2 is not helping someone to comprehend the antimicrobial activity of the essential oils.

Authors: The graph of figure 2 and the legend of the same were altered for a better understanding of the antimicrobial results.

·         Reviewer: Line 227. Table 6 is missing.

Authors: Table 6 does not exist, it was a formatting error. "Table 6" has been corrected to "table 2".

·         Reviewer: Lines 263-293 of the discussion are repeated again in lines 294-324.

Authors:  The Repeated lines were deleted.

·         Reviewer: The discussion should focus in the comparison of the results from this study to those already published.

Authors:   The discussion was altered, focusing on the results and comparing with the literature, as requested for review.

·         Reviewer: Line 403. Your results are not supporting this conclusion.

Authors: The antimicrobial results have been corrected and thus support the completion of the current line 347.

In the attached file is the revised article, the response to the reviewer and the supplementary material

Reviewer 3 Report

Dear authors

In this manuscript (pharmaceuticals-452597), the authors present the assessment of chemical composition of Pogostemon cablin essential oil and they evaluate the larvicidal activity against Aedes aegypti, antioxidant (by DPPH assay), antibacterial activities (against two gram-negative bacteria Pseudomonas aeruginosa and Escherichia coli and a gram-positive bacterium Staphylococcus aureus) and the toxicity against Artemia salina.  

Overall the manuscript has interest, mainly because the larvicidal activity against Aedes aegypti, but need significant improvement.

My major concern has to do with control missing in some experiments and the antibacterial activity results.

The presentation of the antibacterial results is incomprehensible. There seems, to me, to be some error in the construction of the graphs (see comments in detail in the attached document).

Your best result is the larvicidal activity but, compared with what? You didn’t present the LD50 of a well-recognized larvicidal compound to compare and you must to present it. The authors should present the result obtained to the positive control (a standard used as reference).

The table 1 must include all the identified compounds.

On the presentation of the results many other comments are presented in the attached file. Please take them into account in a possible revised version.

The discussion of the results should be much more an analysis of the results obtained in this work compared to the one described in the literature, and not the opposite (exposure of what is reported in the literature without related it to the results obtained).

Reference van Beek and Joulain, (FLAVOUR AND FRAGRANCE JOURNAL, 331, 6-51, 2018) should be include in introduction.

Authors should carefully review the text of the manuscript as it exhibits several word processing problems. Part of the results discussion is repeated (see lines 263-293 which are reprinted in lines 294-324); there are two sections entitled "larvicidal activity"; In the text, (line 227) there is a table 6 that does not exist; The IC50 must be corrected for IC50; The Latin botanical name should be in italic.

Author Response

·      Reviewer: My major concern has to do with control missing in some experiments and the antibacterial activity results.

Authors: Control of all experiments were included in the article. The results of the antimicrobial activity were also corrected.

·      Reviewer: The presentation of the antibacterial results is incomprehensible. There seems, to me, to be some error in the construction of the graphs (see comments in detail in the attached document).

Authors: The graph and legend of Figure 2 were changed for a better understanding of the antimicrobial results.

·      Reviewer: Your best result is the larvicidal activity but, compared with what? You didn’t present the LD50 of a well-recognized larvicidal compound to compare and you must to present it. The authors should present the result obtained to the positive control (a standard used as reference).

Authors: In the article was added the positive control of the larvicide esbiothrin and its LC50.

·      Reviewer: The table 1 must include all the identified compounds.

Author: Table 1 (in the article) includes all compounds that could be identified, with 84.13% of the total present in the sample. Thus, 15.87% was the percentage of unidentified compounds through GC MS analysis.

In the attached file, the Table 1 shows the identified and unidentified compounds, totaling 100%. The respective peaks indicated in the table refer to those shown in the chromatogram of figure 1.

·      Reviewer: On the presentation of the results many other comments are presented in the attached file. Please take them into account in a possible revised version.

Authors: On the attached file, in relation to the results the following changes were made:

Line 211: the data mentioned in this line are already in tebela 1, are the compounds of numbers 6, 14, 17, 24 and 25.

Line 227: the LC50 of the larvicide used as positive control was added. Table 6 was corrected for table 2.

Line 230: subtitle '' larvicidal activity '' has been replaced for "antioxidant activity"

Line 234: The IC50 of the sample was calculated by extrapolation. As suggested, the concentrations tested were specified in the methodology, and the results showed in the text the IC50 values of the sample compared to the IC 50 of the positive control

Line 249: caption of the graph was changed to a better understanding of antimicrobial activity.

Line 254: The MIC and MBC result is 62.5 μg.mL-1. This paragraph has been modified for better understanding.

Line 294: Line 294 to 323 has been deleted because it is a repeating text

·      Reviewer: The discussion of the results should be much more an analysis of the results obtained in this work compared to the one described in the literature, and not the opposite (exposure of what is reported in the literature without related it to the results obtained).

Authors: The discussion was altered as suggested, the results were related to the literature.

·      Reviewer: Reference van Beek and Joulain, (FLAVOUR AND FRAGRANCE JOURNAL, 331, 6-51, 2018) should be include in introduction.

Authors: This reference is included in the article, line 63, reference number 9.

·      Reviewer: Authors should carefully review the text of the manuscript as it exhibits several word processing problems. Part of the results discussion is repeated (see lines 263-293 which are reprinted in lines 294-324); there are two sections entitled "larvicidal activity"; In the text, (line 227) there is a table 6 that does not exist; The IC50 must be corrected for IC50; The Latin botanical name should be in italic.

Authors: All word processing problems have been fixed.

Authors' Note: For the other comments in the attached document:

Line 51-53: the reference Kusuma and and Mahfud (2017) is already mentioned in the introduction

Line 211: the data mentioned in this line are already in table 1

Line 294-324 was deleted because it was repeated (formatting error)

Line 403: The results have been modified and are now in agreement with this conclusion.

The other formatting changes were performed, such as corrections of subscript numbers and names in italics.

In the attached file is the revised article, the response to the reviewer and the supplementary material

Round 2

Reviewer 2 Report

The article has greatly improved.

Author Response

I did not find Round 2 of your comments to correct. I believe that round 1 was enough to clarify doubts and corrections. So, thank you very much for contributing our article.

Reviewer 3 Report

Dear authors

Although the manuscript pharmaceuticals-452597, has been greatly improved, a few minor points remain that need to be improved.

a) The tables are not numbered sequentially (table 2 jump to table 4). Correct also in the text (lines 257, 259, 261).

b) In figure 2, the authors should replace the comma by point in the values presented in the X axis.

c) line 277: main substance instead main substance.

d) lines 229-233: the authors should change the sentences once the essential oil exhibits an LD50 value ~104 folds higher than the positive control. Thus, compared with the positive control the sample is 104 folds less active. On the other hand, “an LD50 below 100 ppm is good” but the essential oil exhibits an LD50 much higher than 100 ppm. So, is the larvicidal activity so relevant as the authors reclaim? Please reformulate the sentences

Author Response

Answer letter

Reviewer: The tables are not numbered sequentially (table 2 jump to table 4). Correct also in the text (lines 257, 259, 261).

Authors: These corrections were made

Reviewer: In figure 2, the authors should replace the comma by point in the values presented in the X axis.

Authors: These corrections were made

Reviewer: line 277: main substance instead main substance.

Authors: These corrections were made

Reviewer: lines 229-233: the authors should change the sentences once the essential oil exhibits an LD50value ~104 folds higher than the positive control. Thus, compared with the positive control the sample is 104 folds less active. On the other hand, “an LD50 below 100 ppm is good” but the essential oil exhibits an LD50 much higher than 100 ppm. So, is the larvicidal activity so relevant as the authors reclaim? Please reformulate the sentences

Authors: 100 ppm is equivalent to 100 μg / mL, then the essential oil LC50 was 28.43 μg /mL, below 100 μg / mL or 100 ppm (line 227). In the article, the value of 100 μg / mL was inserted in parentheses next to 100 ppm for better understanding.
